## [Decision Letter · Decision Letter 0]

31 Jul 2025

PGENETICS-D-25-00523

COG5 deficiency disrupts cellular copper homeostasis and underlies the impaired mitochondrial OXPHOS function

PLOS Genetics

Dear Dr. Lou,

Thank you for submitting your manuscript to PLOS Genetics. After careful consideration, we feel that it has merit but does not fully meet PLOS Genetics's publication criteria as it currently stands. Therefore, we invite you to submit a revised version of the manuscript that addresses the points raised during the review process.

Please submit your revised manuscript within 60 days Sep 29 2025 11:59PM. If you will need more time than this to complete your revisions, please reply to this message or contact the journal office at plosgenetics@plos.org. Please include the following items when submitting your revised manuscript:

We look forward to receiving your revised manuscript.

Kind regards,

Carmen Priolo, M.D., Ph.D.

Academic Editor

PLOS Genetics

Gregory Cooper

Section Editor

PLOS Genetics

Aimée Dudley

Editor-in-Chief

PLOS Genetics

Anne Goriely

Editor-in-Chief

PLOS Genetics

**Journal Requirements:**

At this stage, the following Authors/Authors require contributions: Yuwei Zhou, Xue Ma, Xiaofei Zeng, Keyi Li, Ruowei Zhu, Zhehui Chen, Liqin Jin, Jianxin Lyu, Yanling Yang, and Xiaoting Lou. Please ensure that the full contributions of each author are acknowledged in the "Add/Edit/Remove Authors" section of our submission form.

The list of CRediT author contributions may be found here: https://journals.plos.org/plosgenetics/s/authorship#loc-author-contributions

https://journals.plos.org/plosgenetics/s/submission-guidelines#loc-parts-of-a-submission

Potential Copyright Issues:

- Graphical Abstract. Please confirm whether you drew the images / clip-art within the figure panels by hand. If you did not draw the images, please provide (a) a link to the source of the images or icons and their license / terms of use; or (b) written permission from the copyright holder to publish the images or icons under our CC BY 4.0 license. Alternatively, you may replace the images with open source alternatives. See these open source resources you may use to replace images / clip-art:

6) Please ensure that the funders and grant numbers match between the Financial Disclosure field and the Funding Information tab in your submission form. Note that the funders must be provided in the same order in both places as well.

**Reviewers' comments:**

Reviewer's Responses to Questions

**Comments to the Authors:**

Reviewer #1: Yuwei et al. investigate the role of COG5 in mitochondrial function and identify a novel link between COG5 deficiency and disrupted oxidative phosphorylation (OXPHOS). They show that loss of COG5 leads to intracellular copper overload, impaired iron-sulfur cluster activity, and reduced complex I assembly. These mitochondrial defects were rescued by COG5 re-expression or copper chelation. The identification of a patient with biallelic COG5 variants and a Leigh-like phenotype underscores the clinical relevance of their findings.

Strengths:

-The study addresses a significant gap in understanding COG5 function beyond glycosylation pathways.

-Comprehensive use of COG5 knockout, rescue, and mutant cell models strengthens mechanistic claims.

The identification of a patient with biallelic COG5 variants and Leigh-like syndrome provides clinical interest/relevance.

-Rescue of OXPHOS defects via COG5 re-expression or copper chelation is a compelling therapeutic insight.

The data presentation is overall clear, and the manuscript is generally well written.

Major comments:

1)Authors need to clarify the rationale for selecting copper concentrations used in cell treatments and if they reflect physiological/pathological levels.

2) Most western blot images and blue-native PAGE gels are provided as full-length in the supplementary figures, which is excellent. However, some of the figure panels (e.g., Figures 3F and 5J) could benefit from clearer labels, improved contrast, or better resolution for quantitative interpretation. The legends in some figures are minimal and would benefit from additional explanation (e.g., specifying loading controls and treatment conditions more clearly).

3) While ATP7A downregulation is noted, further discussion of how COG5 may regulate ATP7A trafficking or stability would enhance the mechanistic understanding.

4) Consider commenting on whether similar copper-related mitochondrial dysfunction may be relevant in other CDG subtypes or Golgi disorders.

5) The patient case is compelling; however, more detail on the differential diagnosis or exclusion of other known mitochondrial genes would be helpful.

6)Ensure that all figures have sufficient resolution and legends for standalone interpretation.

7) The use of patient-derived immortalized B cells is a strength, authors should briefly discuss any limitations of this cell model in reflecting in vivo mitochondrial phenotypes.

8) In the Discussion, authors may consider elaborating on whether the copper chelation strategy (e.g., TTM) has translational potential or safety concerns in the context of mitochondrial disease.

9) Authors should clarify if the observed reduction in complex I is due solely to protein abundance or if there are any assembly/stability defects indicated by native gel profiles.

10) The role of ATP7A downregulation is mentioned but not functionally validated or deeply discussed. The broader relevance to other CDGs or mitochondrial diseases is not explored but could be.

Minor comments:

-Line 35: consider revising “deficit” to “deficiency” for consistency.

-Abstract: The phrase “COG5-lacking/revive cell models” is awkward; consider rephrasing as “COG5-deficient and rescue cell models” for clarity.

-Materials & Methods: Ensure consistent use of units (e.g., use “μL” not “ul” and “μg/mL” consistently).

Figure Legends: Check that all acronyms (e.g., TTM, DUB) are defined upon first use in figure legends as well as in the main text.

-While graphs are shown, it is not always clear if significance values (e.g., p < 0.05) are included across all panels. These should be consistently indicated. Error bars are not always labeled as SD or SEM.

-Double-check spacing around units and punctuation throughout for typographical consistency (e.g., “20 μM” not “20μM”; “p < 0.05” with space before and after symbol).

-Model Limitations: The use of HEK293T and EBV-transformed B cells is appropriate but has limitations that are not acknowledged in the discussion.

Reviewer #2: Manuscript ID: PGENETICS-D-25-00523

Type of manuscript: Article

Title: COG5 deficiency disrupts cellular copper homeostasis and underlies the impaired mitochondrial OXPHOS function.

Thanks for the opportunity to review “COG5 deficiency disrupts cellular copper homeostasis and underlies the impaired mitochondrial OXPHOS function.” To date, less than 20 patients have been reported in the literature. As such, the pathogenic and phenotypic spectrum of COG5-related disease(s) is likely incompletely characterized; thus, the potential for new or expanded phenotypes is possible. This manuscript intends to promote the characterization and significance of COG5 in mitochondrial OXPHOS, particularly to complex 1 assembly. This is a well-presented manuscript with extensive functional characterization of COG5 functions, providing mechanistic insights into the role of COG5 beyond Golgi-mediated glycosylation modifications. Without the supplementary data, it would have been impossible to assess the quality of the functional data due to the poor resolution of the figures. However, the patient included in this paper doesn’t have COG5 deficiency, the variants detected in this patient are benign.

Major clarification/modifications:

1. Graphical abstract 1, all figures, needs higher resolution, Without the supplementary data, it would have been impossible to assess the quality of the functional data due to the poor resolution of the figures.

2. For the variants detected in the patient, should provide the protein impact i.e NM_006348.5: c.1826T>C; NP_006339.4: p.I609T in the manuscript.

3. The variants interpretation is incorrect (S1 data): The NM_006348.5: c.1826T>C; NP_006339.4: p.I609T is benign per ACMG variant interpretation guideline. As reported by several laboratory entries in ClinVar https://www.ncbi.nlm.nih.gov/clinvar/variation/225317/. Moreover, there are 11 homozygous in GnomAD v 4 (https://gnomad.broadinstitute.org/variant/7-107248423-A-G?dataset=gnomad_r4)

4. The c.1290C>A (p.F430L) has not been previously reported in affected individuals. This missense is benign per ACMG variant interpretation guideline. As reported by several laboratory entries in ClinVar (https://www.ncbi.nlm.nih.gov/clinvar/variation/358460/). Moreover, there are 11 homozygous in GnomAD v 4 (https://gnomad.broadinstitute.org/variant/7-107298165-G-T?dataset=gnomad_r4)

5. S6 Fig Please provide at the coding and protein impact. “Mutation” should not be used per ACMG guidelines and the HGVS Nomenclature is an internationally-recognized standard for the description of DNA, RNA, and protein sequence variants.

6. Removed all data referring for the patient; this patient doesn’t have COG5 deficiency, both variants detected are benign.

7. Authors can discuss the previously reported patients in the context of the functional studies they have performed.

Reviewer #3: attachment

**Have all data underlying the figures and results presented in the manuscript been provided?**

Reviewer #1: Yes

Reviewer #2: Yes

Reviewer #3: Yes

PLOS authors have the option to publish the peer review history of their article (what does this mean? ). If published, this will include your full peer review and any attached files.

**Do you want your identity to be public for this peer review?** For information about this choice, including consent withdrawal, please see our Privacy Policy .

Reviewer #1: **Yes:** Krinio Giannikou

Reviewer #2: No

Reviewer #3: No

**Figure resubmission:**
---

## [Decision Letter · Decision Letter 1]

19 Nov 2025

PGENETICS-D-25-00523R1

COG5 deficiency disrupts cellular copper homeostasis and underlies the impaired mitochondrial OXPHOS function

PLOS Genetics

Dear Dr. Lou,

Thank you for resubmitting your manuscript to PLOS Genetics. After careful consideration, we feel that the manuscript should be further improved to fully meet PLOS Genetics's publication criteria. Therefore, we invite you to submit a revised version of the manuscript that addresses the points raised during the review process.

Please submit your revised manuscript within by Dec 19 2025 11:59PM. If you will need more time than this to complete your revisions, please reply to this message or contact the journal office at plosgenetics@plos.org. Please include the following items when submitting your revised manuscript:

We look forward to receiving your revised manuscript.

Kind regards,

Carmen Priolo, M.D., Ph.D.

Academic Editor

PLOS Genetics

Gregory Cooper

Section Editor

PLOS Genetics

Aimée Dudley

Editor-in-Chief

PLOS Genetics

Anne Goriely

Editor-in-Chief

PLOS Genetics

**Journal Requirements:**

1) Please update the legends for figures 1, 2B, 4A, and 5J to indicate that they were created using BioRender.

**Reviewers' comments:**

Reviewer's Responses to Questions

**Comments to the Authors:**

Reviewer #3: No more

Reviewer #4: Overview of the strongest claims and significance of the work:

The manuscript by Zhou et al. essentially claims a chain of causality connecting the following events (AB-C etc):

(A) Knock-out of COG5

(B) Decrease in ATP7A protein level

(C) Elevated copper levels

(D) Disrupted iron-sulfur clusters, particularly mitochondrial ones

(E) Reduction in OXPHOS complexes, especially complexes I and III

(F) Mitochondrial disease

The evidence for (A)  (E)/(F) is fairly strong, despite major gaps in some of the intervening logical steps. Proteomic analysis of COG5 knock-out clones, add-backs and parental wild-type cells (Fig. 3) indicate that loss of COG5 results in loss of other COG proteins (as expected) and OXPHOS complexes, particularly I and III. This observation is validated by blue native gels and enzyme assays, which are overall convincing. In Figure 6, patient-derived lines exhibiting a COG5 deficiency also have reduced levels of complex I. The general pattern of the data seem to be that, among the OXPHOS complexes, complex I levels are by far the most sensitive to loss of COG5. The authors cite prior studies characterizing how COG complex defects can disrupt mitochondria, but the specific connection to complex I (and to a lesser extent III) is to my knowledge novel and interesting.

The data supporting (A)  (C) and (C)  (F) also appear to be fairly strong (again despite gaps in intervening logical steps). Figure 4H shows that COG5 knock-out cells have much higher levels of copper. It is curious that copper levels in the add-back are much closer to those in the knock-out than the parental control, and the normalization to cell number is also not ideal because it can be confounded by differences in cell size, which might be significant since the knock-out cells are likely quite sick according to a variety of presented data (e.g., electron microscopy in Fig S1G). However these issues do not really call into question the claim that loss of COG5 results in higher levels of cellular copper, especially given the experiment in Figure 4K, which shows that treatment of the knock-out cells with copper chelator tetrathiomolybdate (TTM) restores the levels of complexes I and III. For technical reasons, the authors were unable to measure copper levels in the patient derived lines analyzed in Figure 6, but there they also show that these cells have profoundly reduced levels of COG5 (Fig. 6B) and have concomitantly reduced levels of Complexes I and III (Fig 6F) and that TTM treatment restores these levels (Fig 6J).

In my view, the most important result of the entire work is that tetrathiomolybdate treatment can rescue the loss of OXPHOS complexes (I and III) resulting from deletion of COG5 (Figures 4K-L, S2D-E). Prior studies may have characterized interactions between COG complex deficits, copper homeostasis and OXPHOS deficiencies, but as far as I can tell, none had demonstrated that a copper chelator can reverse such changes resulting from disruption of the COG complex. Such an experiment strongly supports the claim (A)  (C)  (F). Moreover, the tetrathiomolybdate experiments are key to the medical implications of the work, especially since tetrathiomolybdate is already used clinically. In the Discussion (lines 451-455), the authors explicitly suggest "a potential therapeutic role for copper chelation strategies... agents such as [tetrathiomolybdate]" in treatment of COG5 deficiency.

Given their importance, I have some critical questions about the tetrathiomolybdate experiments. In lines 655-656, the methods state that the dose of tetrathiomolybdate was 20 micromolar, which is a bit high, and might be expected to have significant cytotoxic effects (e.g., inhibition of SOD1). For example, the data presented show reduction in levels of Complex IV, which is a known effect of tetrathiomolybdate treatment. Perhaps for reasons of cytotoxicity, the methods also state that the duration of treatment was only 4 hours. That seems too brief for the observed rescue of Complex I and III assembly, but I cannot be entirely sure. In any case, it seems unusually/unnecessarily short. For example in contrast, the CuCl2 / disulferam treatment was 24 hours (line 655). The authors cite the cuproptosis paper (Tsvetkov et al. Science 2022) for the copper overload treatment protocol (Fig. 4I of the manuscript), but in the cuproptosis paper, in its Fig S1B-C, one can see that even just 1 micromolar tetrathiomolybdate can protect cells from more excess copper than used here in Fig. 4I. It is curious then that such a high dose of tetrathiomolybdate is used to rescue the phenotype of COG5 deficient cells, which were not exposed to excess copper. The authors should provide rationalization for their choice of dose and duration of tetrathiomolybdate treatment. If further experiments are still possible, the paper would be strengthened by including at least a dose curve of TTM and perhaps also a time course.

Overview of the weakest claims and problems in the work:

1) the role of ATP7A

The authors claim, in lines 167-188, that loss of COG5 reduces ATP7A levels and that ATP7A reduction is responsible for the observed elevated levels of cellular copper. The data in figures 4C and 4E are convincing that loss of COG5 can cause almost 2-fold reduction in ATP7A, but the obvious next experiment is to overexpress ATP7A. Why wasn't this done? Otherwise, there is no evidence here that the change in levels of ATP7A are affecting copper concentrations. ATP7A is known to regulate copper homeostasis, but its loss is reported to cause either a decrease or an increase in cellular copper concentrations depending on the context.

Moreover, the co-IP in Figure 4B is provided as demonstration of a physical interaction between COG5 and ATP7A (lines 174-175), but because Fig 4B shows that beta-actin also co-precipitates with COG5, there's actually no evidence here of the claimed interaction. Such an interaction may have been previously reported, but the authors have to rigorously reproduce that result in their system if they wish to make such claims.

The claim that COG5 regulates the half-life of ATP7A (Fig. 4G; lines 180-181) is also not supported. The last time point of the time cycloheximide time course is uninterpretable because it shows a major decrease in beta-actin levels, presumably because the cells are quite sick at this point, and if one only compares the time points for 0-8hrs, it is unclear if there's any difference in half life because the initial/basal levels of ATP7A are lower in the knock-out.

Finally, why weren't ATP7A levels probed in the patient-derived cells?

2) the role iron sulfur cluster (ISC) activity

Lines 233-234 say that "the results showed that COG KO cells showed no significant decrease in mitochondrial ACO-2 in-gel activity compared with control cells." Apparently this is a typo because a significant decrease is claimed elsewhere, such as in the section heading (line 217), but this typo is actually a fairer interpretation of the data. The ACO2 activity bands in Figs 4D, 4F are so faint/overexposed that it is difficult to discern anything. The quantification of these bands in the bar charts of Figs 4E, 4G is unconvincing. With in-gel activity assays, a real effect should be apparent by eye or else it is not trustworthy.

Moreover, even if a decrease in iron sulfur cluster activity were real in these cells, there are so many other ways one could explain it. For example, iron sulfur clusters are sensitive to reactive oxygen species (ROS), so the reduction might be attributable to the elevated ROS levels shown in Fig 3G. Those elevated ROS levels might be a result of OXPHOS complex imbalance, rather than a cause.

If the authors wish to claim that disruption of iron sulfur activity (ISC) is part of the mechanism of how COG5 deficiency affects OXPHOS, then effect sizes in the ISC activity assays need to be larger and clearer, and crucially, they should be reversed by the treatment that rescues the OXPHOS complexes, i.e., tetrathiomolybdate.

Relatively minor issues/suggestions:

1) Fig. 2F is illegible, even in the TIF file, and its caption ("This chord diagram showcases leading-edge proteins, highlighting those significantly driving the selected GSEA pathways upregulated by exercise to unveil potential signaling pathways") is apparently copied from Figure 5 of a 2022 Frontiers in Immunology paper (https://pubmed.ncbi.nlm.nih.gov/36189306).

2) There are no OXPHOS-related data in Figure 2 but its caption is "proteomics analysis indicates that COG5 deficiency results in impaired oxphos." That caption really pertains to Figure 3.

3) Immunoblots should indicate the specific proteins probed. There are no antibodies for complex I, II etc. There are antibodies for specific proteins in those complexes and those should be written in the figure. I'm assuming this information is in Table S1, which the text says contains the list of antibodies, but I was not able to access it.

4) the authors could consider the possibility of a genetic interaction between the COG5 mutations in the patient case study. Prior reviewers raised the concern that the COG5 mutations in the patient should be benign because databases indicate that there exist asymptomatic individuals homozygous for those mutations (the Discussion in lines 436-440 wrestles with this "conflicting evidence" issue), but the patient in question is a compound heterozygote. The term intragenic or allelic or inter-allelic complementation describes the phenomenon in which a compound heterozygous mutations has a qualitatively different phenotype from individuals homozygous for either mutation, and formally speaking, such a possibility would explain the apparent contradictions among the evidence here. This might be testable by expressing both of the COG5 F430L and I609T mutants together in the same knock-out cells and comparing the resulting phenotypes to the situation in which the mutants are expressed separately.

**Have all data underlying the figures and results presented in the manuscript been provided?**

Reviewer #3: None

Reviewer #4: **No:** I couldn't find a spreadsheet with the proteomics data or table S1 (antibodies used)

PLOS authors have the option to publish the peer review history of their article (what does this mean? ). If published, this will include your full peer review and any attached files.

**Do you want your identity to be public for this peer review?** For information about this choice, including consent withdrawal, please see our Privacy Policy .

Reviewer #3: No

Reviewer #4: No

**Figure resubmission:**
---

## [Editor Report · Decision Letter 2]

2 Mar 2026

Dear Dr Lou,

We are pleased to inform you that your manuscript entitled "COG5 deficiency disrupts cellular copper homeostasis and underlies the impaired mitochondrial OXPHOS function" has been editorially accepted for publication in PLOS Genetics. Congratulations!

Yours sincerely,

Carmen Priolo, M.D., Ph.D.

Academic Editor

PLOS Genetics

Gregory Cooper

Section Editor

PLOS Genetics

Aimée Dudley

Editor-in-Chief

PLOS Genetics

Anne Goriely

Editor-in-Chief

PLOS Genetics

BlueSky: @plos.bsky.social

Comments from the reviewers (if applicable):

**Data Deposition**

http://datadryad.org/submit?journalID=pgenetics&manu=PGENETICS-D-25-00523R2

**Press Queries**

---

## [Editor Report · Acceptance letter]

PGENETICS-D-25-00523R2

COG5 deficiency disrupts cellular copper homeostasis and underlies the impaired mitochondrial OXPHOS function

Dear Dr Lou,

We are pleased to inform you that your manuscript entitled "COG5 deficiency disrupts cellular copper homeostasis and underlies the impaired mitochondrial OXPHOS function" has been formally accepted for publication in PLOS Genetics! Your manuscript is now with our production department and you will be notified of the publication date in due course.

With kind regards,

Anita Estes

PLOS Genetics

On behalf of:
